# Surface Functionalization with Polyethylene Glycol and Polyethyleneimine Improves the Performance of Graphene-Based Materials for Safe and Efficient Intracellular Delivery by Laser-Induced Photoporation

**DOI:** 10.3390/ijms21041540

**Published:** 2020-02-24

**Authors:** Jing Liu, Chengnan Li, Toon Brans, Aranit Harizaj, Shana Van de Steene, Thomas De Beer, Stefaan De Smedt, Sabine Szunerits, Rabah Boukherroub, Ranhua Xiong, Kevin Braeckmans

**Affiliations:** 1Laboratory of General Biochemistry and Physical Pharmacy, Faculty of Pharmaceutical Sciences, Ghent University, B-9000 Ghent, Belgium; jingli.liu@ugent.be (J.L.); toon.brans@ugent.be (T.B.); Aranit.Harizaj@UGent.be (A.H.); Stefaan.DeSmedt@UGent.be (S.D.S.); ranhua.xiong@ugent.be (R.X.); 2University Lille, CNRS, Centrale Lille, ISEN, University Valenciennes, UMR 8520-IEMN, F-59000 Lille, France; neu_lcn@163.com (C.L.); sabine.szunerits@univ-lille.fr (S.S.); rabah.boukherroub@univ-lille.fr (R.B.); 3Laboratory of Pharmaceutical Process Analytical Technology, Faculty of Pharmaceutical Sciences, Ghent University, B-9000 Ghent, BelgiumThomas.DeBeer@UGent.be (T.D.B.); 4Centre for Advanced Light Microscopy, Ghent University, B-9000 Ghent, Belgium; 5Joint Laboratory of Advanced Biomedical Technology (NFU-UGent), College of Chemical Engineering, Nanjing Forestry University (NFU), Nanjing 210037, China

**Keywords:** photoporation, intracellular delivery, nanomaterial functionalization, graphene-based materials, colloidal stability

## Abstract

Nanoparticle mediated laser-induced photoporation is a physical cell membrane disruption approach to directly deliver extrinsic molecules into living cells, which is particularly promising in applications for both adherent and suspension cells. In this work, we explored surface modifications of graphene quantum dots (GQD) and reduced graphene oxide (rGO) with polyethylene glycol (PEG) and polyethyleneimine (PEI) to enhance colloidal stability while retaining photoporation functionality. After photoporation with FITC-dextran 10 kDa (FD10), the percentage of positive HeLa cells (81% for GQD-PEG, 74% for rGO-PEG and 90% for rGO-PEI) increased approximately two-fold compared to the bare nanomaterials. While for Jurkat suspension cells, the photoporation efficiency with polymer-modified graphene-based nanomaterial reached as high as 80%. Cell viability was >80% in all these cases. In addition, polymer functionalization proved to be beneficial for the delivery of larger macromolecules (FD70 and FD500) as well. Finally, we show that rGO is suitable for photoporation using a near-infrared laser to reach 80% FD10 positive HeLa cells at 80% cell viability. We conclude that modification of graphene-based nanoparticles with PEG and especially PEI provide better colloidal stability in cell medium, resulting in more uniform transfection and overall increased efficiency.

## 1. Introduction

Physical transfection methods are in high demand for the intracellular delivery of extrinsic molecules, especially in an in vitro or ex vivo context [1]. One example is the intracellular delivery of biomolecules, such as siRNA, mRNA, pDNA, or CRISPR/CAS9 ribonucleoprotein complexes. Importantly, the delivery efficiency of these molecules interferes with gene expression, for instance, to when attempting to unravel molecular pathways and identify new drug targets [2,3]. Genetic engineering of cells is also of importance to produce more potent cells for therapeutic purposes, such as the production of engineered immune cells for anti-cancer therapy [4,5]. Another example is the intracellular delivery of contrast agents, for instance for microscopic imaging or for in vivo cell tracking [6,7]. However, the cell membrane is a natural barrier preventing spontaneous intracellular entry of most macromolecules. While a plethora of chemical transfection agents has emerged over the years, they mostly rely on endocytic uptake, so that the compounds are inside endosomes rather than the cytosol. Unfortunately, efficient escape from endosomes into the cytosol is still one of the major bottlenecks in the drug delivery field today, so many chemical transfection agents do not perform well in combination with hard-to-transfect primary cells [8,9,10,11]. This is why there is intensive research ongoing on improved physical transfection technologies, which are based on the use of physical forces to temporarily increase the permeability of the cell membrane, allowing extrinsic compounds to permeate from the surrounding cell medium directly into the cytosol.

At present, electroporation is the oldest but still the most often used physical transfection method as it is readily commercially available from several manufacturers. With carefully tuned electrical pulses, it allows the transfection of a large amount of cells, the downside being that it is often associated with high levels of acute toxicity [12]. Microfluidic cell squeezing is a recently developed more gentle approach based on flowing cells through narrow constrictions in a microfluidic channel. Thus the shear forces induce pores in the cell membrane. It reaches a very good delivery efficiency with high cell viability, but with the drawback that it can only be applied to cells in suspension. While optimization of the microfluidic device is very much cell size-dependent [13,14]. Another recent more gentle physical transfection technology is the laser-induced photoporation that conveniently can be applied to both adherent and suspension cells [5,15,16]. Cells are incubated with light-sensitive nanoparticles (NPs) that can bind to the cell membrane and locally induce enhanced permeation by photothermal effects upon laser irradiation [17]. When pico- to nanosecond laser pulses are used, one can make use of essentially two distinct photothermal effects to enhance membrane permeability. At low laser pulse fluences, the permeabilizing effect is mostly due to local heat generation [18], while at higher laser fluences, water vapor nanobubbles (VNBs) can form around the NP surface. When the VNBs collapse, the physical force can induce transient pores in the cell membrane [17,19]. Spherical gold nanoparticles (AuNP) of 50–100 nm diameter have been used the most as sensitizers for photoporation due to their excellent photothermal properties [20]. However, upon pulsed laser irradiation, AuNP tends to fragment into very small particles which raises concerns about safety, especially in the context of the production of therapeutic engineered cells [21]. Indeed, very small gold nanoparticles have been shown to be able to intercalate into DNA, potentially inducing genotoxic effects [22]. There are also practical constraints when using AuNP whose plasmonic absorption peak very much depends on the particle size and shape [23], so that different laser wavelengths require the optimization of different types of AuNP and vice versa.

Recently, we have demonstrated that graphene-based nanoparticles represent an interesting alternative to AuNP as sensitizing nanoparticles for photoporation. Graphene-based materials, such as reduced graphene oxide (rGO), have been widely used for biomedical applications in the past few years due to several interesting physicochemical properties, such as having a large specific surface area and outstanding photothermal properties combined with a broad light absorption spectrum [24,25,26]. In particular, we have demonstrated that graphene quantum dots (GQDs) were excellent sensitizing nanoparticles to controllably deliver fluorescent contrast agents into cells by photoporation [27]. However, for bare GQD, it was noted that upon addition to cell cultures they aggregated quickly, leading to the formation of large and energetic water VNBs upon laser irradiation, thus inducing more toxicity as compared to traditionally used AuNP.

To further improve the efficiency and safety of graphene-based materials for photoporation, we here propose surface modification with polymers to enhance their colloidal stability. This should reduce cytotoxicity during photoporation while we aim to keep the same delivery efficiency. Both GQD and reduced rGO NPs are included in this study, as two distinct types of graphene-based materials. GQD is included as a benchmark with our previous work, while rGO is additionally included since it has a broader absorption spectrum stretching into the near-infrared (NIR) range, thus offering even more flexibility regarding the use of laser wavelengths. Both GQD and rGO were modified with polyethylene glycol (PEG) as a common strategy to enhance the colloidal stability of nanoparticles. In addition, since cell attachment is usually better with cationic nanoparticles, rGO was functionalized with polyethyleneimine (PEI) as well to investigate if this has a net positive effect on the photoporation efficiency without compromising colloidal stability or cell viability. GQD and rGO were characterized in terms of size, zeta-potential, surface functionalization, and laser fluence threshold for VNB formation. Next, their performance as sensitizers for photoporation with 561 nm nanosecond pulsed laser irradiation was tested on adherent cells (HeLa) and suspension cells (Jurkat). The delivery efficiency of FITC-dextran of different molecular weights is optimized as a function of nanoparticle concentration and cell viability. Thanks to their broader absorption spectrum, the concept of NIR photoporation at 800 nm was specifically tested with rGO-PEI as well. This is of interest to explore the future use of photoporation for intracellular delivery in thick biological tissues, where NIR laser light has better penetration depth [28,29].

## 2. Results and Discussion

### 2.1. Physicochemical Characterization of GQD and rGO Before and after Surface Functionalization

As schematically shown in Figure 1, both GQD and rGO are modified with PEG as the most common strategy to enhance the colloidal stability of nanoparticles in bio-fluids (such as cell medium). rGO is additionally modified with branched PEI (25 kDa) to specifically evaluate the benefit of a cationic coating on cell attachment and photoporation efficiency vs. cytotoxicity. PEG is coupled to GQD and rGO through a covalent bond between the terminal amino group on PEG and carboxyl groups on the nanoparticles. Unbound PEG is removed by dialysis. The functionalization of rGO with PEI is simply based on electrostatic adsorption between the strongly negatively charged rGO and cationic PEI molecules. Unbound PEI is removed by repeated centrifugation and washing to avoid toxicity to cells by free PEI [30].

The extinction spectra of GQD and GQD-PEG are shown in Figure 2a. Since there is almost no absorption of the PEG in the range of 200 nm to the NIR wavelength, the absorption spectrum of GQD-PEG was almost the same as that of GQD. The extinction spectrum in Figure 2a confirms that rGO can absorb light over a broader range than GQD, stretching further into the NIR range. After PEGylation, the hydrodynamic size (number mean) in deionized water of GQD was significantly decreased from 45.76 nm to 28.2 nm (Figure 2b). The zeta potential, on the other hand, significantly increased from −27.4 mV to −14.1 mV, confirming the successful conjugation of PEG to the surface of GQD. For rGO, the hydrodynamic size in deionized water significantly decreases and zeta potential significantly increases after PEGylation from 320 nm and −52.6 mV to 241 nm and −15.6 mV (Figure 2c), confirming the surface functionalization. The hydrodynamic size and zeta potential of rGO-PEI significantly increase further to 266 nm and +52.5 mV, respectively (Figure 2c). We attribute this to the better colloidal stability after surface functionalization, meaning that GQD-PEG, rGO-PEG and rGO-PEI are more monodispersed as compared to the non-functionalized nanomaterials. The hydrodynamic sizes of GQD and GQD-PEG are also measured in HEPES buffer. As shown in Appendix A, the hydrodynamic size of GQD increased to 101 nm while this was not the case for GQD-PEG (44.2 nm). This clearly indicates aggregation of GQDs. The actual size and morphology of the graphene-based nanomaterials were characterized by Scanning Electron Microscopy (SEM). Comparing Figure 2d,e, GQDs had the tendency to form aggregates on the substrate during the sample preparation, while after PEGylation the GQDs-PEG remained homogeneously distributed on the substrate. The size of individual GQD were slightly larger than that of GQD-PEG with an average value of 36.1 ± 6.2 nm and 29.9 ± 6.0 nm, respectively. A similar tendency to aggregate was observed for unmodified rGO as well (Figure 2f). After functionalized with PEG or PEI, the rGO-PEG or rGO-PEI remained more mono-dispersed (Figure 2g,h). The average size decreased from 348 ± 104 nm to 207 ± 60 nm and 282 ± 87 nm after functionalization with PEG or PEI, respectively. Again this was consistent with the DLS characterization. Taken together, we could conclude that the graphene-based materials were successfully functionalized with polymers and therefore have improved colloidal stability from both DLS measurement and SEM images. Raman spectra were additionally recorded to get further confirmation of successful particle functionalization (Appendix A)). The characteristic D (~1326 cm^−1^) and G (~1603 cm^−1^) bands typical for graphene were present in all spectra. The ratio of peak intensities I_D_/I_G_ was calculated as shown in Appendix A. The ratio of I_D_/I_G_ decreased in all cases after functionalization, due to the polymers shielding the defects [31]. Therefore, the Raman spectra give an additional confirmation that both GQD and rGO were successfully modified with the polymers.

Next, the formation number of VNB is determined as a function of the applied laser pulse fluence (7 ns, 561 nm) by darkfield microscopy. As VNB efficiently scatter light, they can be easily detected and counted in dark field microscopy images (Appendix A) [27,32]. The number of VNBs is counted upon irradiation with a single laser pulse at different laser fluences (Appendix A). From a Boltzmann fit to the data, the so-called VNB threshold could be determined, which is commonly defined as the laser pulse fluence at which 90% of the particles in the irradiated area generate VNBs. For both GQD and rGO, it was found that the VNB threshold increases after functionalization with the polymers (Appendix A). This may be due to the fact that bare graphene-based materials tend to aggregate in PBS, which can generate VNBs at lower energies [33]. Another possible explanation is that this was due to reduced heat transfer from the particles to the surrounding liquid (cell medium) due to the presence of the polymer coating so that more energy is needed to form VNBs. In any case, indirectly, this was another confirmation that the functionalization of particles was successful.

### 2.2. Photoporation of Adherent Cells with Polymer-Modified Graphene-Based Nanoparticles

The intracellular delivery efficiency and cell viability after photoporation is determined for the various types of nanoparticles. FITC-dextran 10 kDa (FD10) was used as a fluorescent model molecule to easily quantify delivery by microscopy and flow cytometry. Photoporation was performed at a laser fluence of twice the VNB threshold, as determined above, to ensure effective VNB formation. The final concentration of the graphene-based nanomaterials was determined by Nanoparticle Tracking Analysis described in the experimental section. Cells were first incubated for 30 min with nanoparticles, after which unbound particles were washed away. Next, a fresh cell medium was added supplemented with FD10, after which laser treatment was applied. As illustrated in Figure 3a, the pulsed laser beam was scanned across the sample so that each cell received a single laser pulse (or two in the overlapping areas). Transient pores were generated in the cell membrane by the VNBs, allowing the FITC-dextran molecules to enter the cytosol. After laser treatment, FD10 was immediately washed away to limit endocytic uptake after which fresh medium was supplemented again. Finally, FD10 uptake was quantified by flow cytometry (percentage of positive cells and their mean fluorescence intensity (MFI)), while cytotoxicity was measured in parallel with the luminescent cell viability assay CellTiter-Glo^®^. As can be seen in Figure 3, in all cases, both the percentage of transfected cells as well as the relative MFI (rMFI) of the positive cells increased with increasing particle concentrations. The rMFI is expressed relative to the control, which are cells incubated with FD10 for the same time as the photoporation procedure but without the addition of nanoparticles or laser exposure. As the transfection efficiency increases, the cell viability gradually decreased. Taking 80% cell viability as a commonly acceptable limit for toxicity, the data in Figure 3b show that in the case of bare GQD, no more than 50% of positive cells could be obtained. This is in line with our previous study that, due to aggregation of bare GQD in cell medium, large VNBs are formed during laser treatment which causes too much damage to cells, limiting the delivery efficiency. [27] Importantly, however, after PEGylation these detrimental effects could be strongly reduced, resulting in photoporation efficiencies as high as 80% with ~80% cell viability and substantially higher MFI values (Figure 3c). We attribute this to the stabilizing action of PEG molecules which renders GQD-PEG more stable in cell medium, thus avoiding aggregation and ensuring that a more monodisperse set of particles can bind to the cell membrane. Exemplary confocal images after photoporation with FD10 are shown in Appendix A, which visually confirm the flow cytometry results.

Similar experiments are performed with rGO, rGO-PEG, and rGO-PEI. For bare rGO again, no more than ~50% of FD10 positive cells could be obtained while keeping >80% viable cells (Figure 3d). After PEGylation, this could be improved to ~75% positive cells for >80% cell viability, although the rMFI decreased slightly from 100 to 74 (Figure 3e). With rGO-PEI, being strongly positively charged, ~90% positive cells were obtained with cell viability close to 90% (Figure 3f). The corresponding rMFI was 127, which was the best result for all tested particles. These results show that functionalization with PEI provided even better results on adherent cells as compared to PEGylation. Exemplary confocal images after photoporation with FD10 are shown in Appendix A. Taken together, we conclude that intracellular delivery efficiency in adherent HeLa cells can be markedly improved after modifying the graphene-based materials with PEG or PEI. rGO-PEG gave similar transfection results as GQD-PEG, although at lower particle concentrations likely due to rGO-PEG being ~10 times larger in size. From all tested particles, rGO-PEI gives the highest percentage of transfected cells as well as rMFI values.

### 2.3. Photoporation of Suspension Cells with Polymer-Modified Graphene-Based Nanoparticles

Given these positive results of functionalized particles on adherent cells, we continued to explore their performance on suspension cells as well (Figure 4a). Jurkat cells were selected, which are a human T cell leukemia cell line that is commonly used as a model for hard-to-transfect primary T cells. The protocol was similar for HeLa cells, with the only difference that cells were collected by centrifugation after every washing step. Flow cytometry results for the delivery of FD10 with increasing particle concentrations are shown in Figure 4b–d, together with cell viability measured by CellTiter-Glo^®^. Exemplary confocal images are shown in Appendix A. Setting 80% cell viability as a threshold again, GQD-PEG gave 56% positive cells with 54 rMFI, as compared to 63% positive cells and 27 rFMI for rGO-PEG, while 80% positive cells and 40 rFMI were obtained with rGO-PEI. Again rGO-PEI gave the highest percentage of transfected cells, while this time the highest rMFI values were obtained by GQD-PEG.

### 2.4. Exploring Intracellular Delivery of Different Sizes of Cargos

Thus far, we had assessed photoporation efficiency with FITC-dextran of 10 kDa, which is representative of smaller nucleic acids such as siRNA [5]. To assess the ability of delivering even larger macromolecules in cells by photoporation with graphene-based materials, we here extend the evaluation to include FITC-dextran of 70 kDa (FD70) and 500 kDa (FD500), which are representative of proteins and larger nucleic acids like mRNA. Photoporation was performed on HeLa cells with GQD and GQD-PEG according to the above optimized conditions (3.56 × 10^9^ nps/mL for GQD and 5.1 × 10^9^ nps/mL for GQD-PEG). The percentage of positive cells is shown in Figure 5a with the corresponding rMFI values in Figure 5b, while representative confocal images are depicted in Appendix A. As expected, the transfection efficiency decreased for larger molecules likely due to a combination of increasing steric hindrance at the membrane pores and slower diffusion. Similar to what was observed before for FD10, the percentage of positive cells was higher for FD70 and FD500 when using GQD-PEG as compared to the bare GQD nanoparticles, although the effect was not statistically significant for FD500. The opposite effect is observed for the rMFI values, which are all higher for bare GQD. This points to the fact that the membrane pores form by GQD-PEG are likely smaller than for bare GQD. This can be explained by aggregation of bare GQD particles upon addition to cell medium, which may lead to bigger VNB and hence larger membrane pores through which larger macromolecules can pass more easily. Of course, this is balanced by higher toxicity as well, as noted before.

The same experiment was repeated for bare rGO in comparison with rGO-PEG and rGO-PEI, according to the above optimized conditions (1.31 × 10^9^ nps/mL for rGO, 0.82 × 10^9^ nps/mL for rGO-PEG, and 12.8 × 10^9^ nps/mL for rGO-PEI). Again the percentage of positive cells was higher for the polymer-modified rGO particles compared to bare rGO (Figure 5c). For both FD70 and FD500, rGO-PEI gave significantly more positive cells. It showed similar results as the rMFI of GQD and GQD-PEG in that the cells that were treated with rGO had significantly higher rMFI than rGO-PEG (Figure 5d). We found the rMFI of FD10 and FD70 for rGO-PEI was slightly higher than that for the rGO, this is because of the highly positive charge of rGO-PEI that could induce more internalization of rGO-PEI with the cell membrane. The fact that this was not seen for FD500 likely means that the pores formed by either of these particles were too small for efficient influx of FD500 so that there was no net beneficial effect. Overall, we conclude that PEI modification of rGO gives the best performance improvement over a wide range of molecular weights.

### 2.5. Intracellular Delivery by NIR Laser Induced-Photoporation with rGO-PEI

As already pointed out, a particular advantage of rGO over GQD is its relatively higher absorbance over a broad range of wavelengths, even stretching into the NIR region. Not only does this provide more flexibility in the laser wavelengths that can be used for photoporation, but it also offers better light penetration depth in biological tissues [34]. With a view on potential future biomedical applications in thicker tissues, we therefore evaluated if efficient photoporation could still be achieved with NIR pulsed laser light. In particular, we used 800 nm laser pulses with a 2 ps pulse width to irradiate rGO-PEI and deliver FD10 into HeLa cells, following the same procedure as before (Figure 6a). Since darkfield microscopy was not available on the photoporation setup with the 800 nm laser, we directly screened the photoporation efficiency as a function of laser pulse fluence. In the absence of laser irradiation (light grey bars in Figure 6b), almost none of the cells had any detectable FD10 signal irrespective of the concentration of rGO-PEI particles. On the other hand, the picosecond laser pulses already slightly permeabilized the cells for FD10 even in the absence of rGO-PEI, reaching ~20% positive cells for the highest laser intensity. This was also visually confirmed by confocal microscopy images in Figure 6c (first column). Consequently, we observed a small decrease in cell viability by the laser alone, decreasing by 16% for the highest laser intensity. In the presence of rGO-PEI; however, laser treatment markedly increased cell transfection efficiency, reaching 80% with cell viability of 80% for a 1.28 × 10^9^ nps/mL and a laser fluence of 0.89 J/cm^2^. At this condition, an rMFI value of 114 was obtained (Appendix A), which is quite similar to that recorded before with 561 nm 7 ns laser pulses. Exemplary confocal microscopy images in Figure 6c confirm the trends observed by flow cytometry, showing more and brighter fluorescent cells with increasing concentrations of rGO-PEI and increasing laser fluence. Together, the results indicate that rGO-PEI can be equally efficiently used for photoporation at NIR wavelengths, making those particles interesting and flexible candidates for applications in thicker biological tissues.

Since its discovery in 2004, graphene is quickly being used in a range of biomedical applications due to its particular physicochemical and optical properties. As a two-dimensional material, the family of graphene-based materials has a high surface area to volume ratio and offers flexible surface functionalization options with polyaromatic structures, as well as hydrophilic regions, being present on the surface [35,36]. These properties offer plenty of opportunities to tune their surface properties, depending on the application at hand. In previous work, we already demonstrated the suitability of bare graphene-based nanoparticles to adhere to cells and form pores in the cell membrane through water VNB formation upon laser irradiation. However, bare nanoparticles are found to partly aggregate upon addition to cell medium, which can be explained by the interaction with physiological components, such as salts, ions and biomolecules, leading to an altered size, shape, and surface chemistry [37,38]. During photoporation, large aggregates induce high local cytotoxicity due to strong photothermal effects. Therefore, here we evaluate polymer surface functionalizations of graphene-based nanoparticles in order to improve their colloidal stability and produce improved photoporation results. PEG is chosen on the one hand as a commonly used hydrophilic polymer to enhance the colloidal stability of nanoparticles [39,40]. However, PEGylation may also reduce contact with cells, which is why we additionally included functionalization with polyethyleneimine (PEI) as a polymer with a strong positive charge for better interaction with the negatively charged cell surface.

We included GQD in our study, of which we recently showed that it can be used to deliver contrast agents into cells by photoporation. In addition, we included rGO as well, because it has higher absorbance in the Vis-NIR region, thus offering excellent compatibility with many laser sources and being of interest for future exploration of photoporation for intracellular delivery into thick tissues, such as organoids or other biofabricated tissues [41,42]. This is quite different from traditionally used plasmonic gold nanoparticles, which only absorb well in a narrow region around the plasmonic peak.

Overall, our results demonstrate that modification of GQD and rGO with PEG or PEI offers a viable route towards higher rates of transfected cells by photoporation for macromolecules at least up to 70 kDa. With FITC-dextran 10 kDa, the percentage of positive HeLa cells increased from 45% for GQD and 48% for rGO to more than 80% for GQD-PEG, 74% for rGO-PEG and more than 90% for rGO-PEI, all with >80% cell viability. In Jurkat suspension cells, as a model for hard-to-transfect human T cells, the percentage of positive cells reached 56% for GQD-PEG, 63% for rGO-PEG, and 80% for rGO-PEI, with 80% cell viability. Generally, the amount of cargo per cell remained equally high independent of polymer functionalization, the only exception being GQD-PEG, which showed a slight reduction in the amount of FD70 and FD500 molecules delivered per cell. We conclude that the polymer functionalized GQD and rGO have greatly improved the intracellular delivery efficiency by laser-induced photoporation, especially with the positively charged PEI. rGO-PEI is, in addition, it was proven to give equally good results in combination with NIR laser excitation.

## 3. Materials and Methods

### 3.1. Cell Culture

HeLa cells (ATCC^®^ CCL-2™) were cultured in DMEM/F-12 (Gibco-Invitrogen, Renfrew Renfrewshire PA4 9RF, UK) supplemented with 10% heat-inactivated fetal bovine serum (FBS) (Biowest, Nuaillé, France), 2 mM glutamine (Gibco-Invitrogen), and 100 U/mL penicillin/streptomycin (Gibco-Invitrogen). Cells were passaged using DPBS (Gibco-Invitrogen, Renfrewshire PA4 9RF, UK) and trypsin-EDTA (0.25%, Gibco-Invitrogen). HeLa cells were cultivated in a humidified tissue culture incubator at 37 °C and 5% CO_2_. All cell culture products were purchased from Life Technologies (Renfrewshire PA4 9RF, UK), unless specifically stated otherwise.

### 3.2. GQD and GQD-PEG Synthesis Method

The synthesis of GQDs was performed according to a previously reported procedure [27]. In Brief, GO was synthesized by following a modified Hummer’s method. rGO was synthesized by the treatment of GO with hydrazine monohydrate. To synthesize small GQD nanoparticles, 100 mg of rGO powder was dispersed in 100 mL of 30% H_2_O_2_ and ultrasonicated for 30 min. The obtained uniformly dispersed solution was kept refluxing for 12 h at 60 °C. The resulting solution was filtered with a 0.2 µm filter to separate from porous reduced graphene. The obtained GQD suspension was further dialyzed with a dialysis cassette 2 kDa MWCO (Thermo Scientific, Waltham, MA, USA ) to remove excess H_2_O_2_ and to separate rGO from small-sized GQDs. The purified GQDs were dissolved in DI water to a final concentration of 1 mg/mL.

For functionalization with PEG, 10 mg N-(3-dimethylaminopropyl)-N’-ethylcarbodiimide hydrochloride (EDC) (Sigma, Overijse, Belgium) and 11.3 mg *N*-hydroxysulfosuccinimide sodium salt (Sulfo-NHS) (Sigma, Overijse, Belgium) were added into 10 mL of the newly synthesized GQDs. The solution was stirred at room temperature for 30 min. Afterward, 20 mg of mPEG-amine (2 kDa MW) (Creative PEGWorks, Chapel Hill, NC, USA) was added underwater bath sonication (BRANSON 2510) (Merck, Overijse, Belgium) at room temperature for 2h. Ten microliters of mercaptoethanol was added into the mixture to quench the reaction. The solution was then transferred into the 14 kDa molecular weight cut-off dialysis membrane (Thermo Scientific, Belgium) and dialyzed in deionized water for two days.

### 3.3. rGO, rGO-PEG and rGO-PEI Synthesis Method

rGO was first synthesized from commercial GO powder. Fifty micrograms of GO (GRAPHITENE, Stevenage SG1 2FX, UK) was dissolved in 50 mL deionized water in a flask. Into the solution, 1.4 g NaOH and 1 g ClCH_2_-COOH were added. Then the flask was put in a sonication water bath (35 kHz) at 80 °C for 2 h. 3 mL of 20% HCl solution was added into the flask to quench the reaction. The rGO was separated by centrifugation at 4600 rpm for 30 min. The supernatant was discarded, and the rGO-CH_2_-COOH was resuspended in deionized water. The centrifugation step was repeated several times to obtain a suspension with a neutral pH [43].

For PEGylation, the newly synthesized rGO was suspended in 50 mL water by bath sonication. The solution was again transferred into a round-bottom flask to which 100 mg mPEG-Amine (2 kD MW, Creative PEGWorks, USA) and 31.25 mg N-(3-dimethylaminopropyl)-N’-ethylcarbodiimide hydrochloride (EDC, Sigma, Begium) were added. The flask was sonicated in the water bath at room temperature for 2 h. Thirty microliters of mercaptoethanol was added into the solution to quench the reaction. The solution was then transferred into a dialysis tube (12–14 kDa MWCO, Dialysis tubing cellulose membrane, Sigma, Belgium) for two days to remove unbound PEG.

For functionalization with PEI, 5 mL of the fresh synthesized rGO-CH_2_-COOH solution was added into a 10 mL round-bottom flask. PEI (branched, ~25 kDa, Sigma, Belgium) was first diluted into deionized water to a 10% (*w/w*) solution. Twenty five milliliters of the 10% PEI solution was gradually added into the flask while standing in the bath sonicator. The flask was first sonicated for 2 h at room temperature and then stirred overnight. The solution was centrifuged at 10,000 rpm for 30 min to remove any remaining free PEI. The washing step was repeated for 3–5 times.

### 3.4. Ultraviolet (UV)-NIR Spectrophotometry, DLS and NanoSight Measurements

UV-NIR distinction spectra were measured on a LAMBDA 950 UV/Vis Spectrophotometer (PerkinElmer, Zaventem, Belgium). Hydrodynamic size and zeta potential were determined by dynamic light scattering (DLS) on a Zetasizer Nano ZS (Malvern, UK). The particle concentration of graphene-based materials was measured on NanoSight (NanoSight LM10, Malvern, UK) in reflection mode.

### 3.5. SEM Imaging

SEM images were obtained by using an FEI Quanta 200F (Thermo Scientific) with an accelerating voltage of 15 kV. GQD, GQD-PEG and rGO were dried on coverslips and coated with a gold layer. rGO-PEG and rGO-PEI were dried on silicon wafers.

### 3.6. Raman Measurement

The measurements were performed with a Raman Rxn1 Microprobe (Kaiser Optical Systems, Ann Arbor, MI, USA) (microscope objective—10×) coupled to a Raman Rxn1 Analyzer (air-cooled CCD detector) via a proprietary holographic optical module. A 785 nm Invictus laser with a power of 400 mW was employed for excitation, and 1 s exposure time was used for the acquisition of spectra.

### 3.7. Laser-Induced Photoporation

Two home-built photoporation setups were used in this work. One was used for photoporation with 561 nm nanosecond pulsed laser light, while the other was used for photoporation with 800 nm picosecond laser light. All experiments were performed with the former, except for a limited set of experiments carried out with the latter, as indicated in the main text.

For the 561 nm laser setup, a 7-ns pulsed laser tuned to a wavelength of 561 nm (Opolette HE 355 LD, OPOTEK Inc., Carlsbad, CA, USA) was used to irradiate the samples. As the photoporation laser beam has a diameter of 150 µm, a scanning procedure was used to treat all cells within each well. The sample was scanned through the photoporation laser beam (20 Hz pulse frequency) using an electronic microscope stage (HLD117, Cambridge, UK). The scanning speed was 2.1 mm/s, and the distance between subsequent lines was 0.1 mm to ensure that each cell received a single laser pulse (or two in the overlapping regions).

The 800 nm laser operates with 2 ps pulses generated at 1 kHz pulse repetition frequency using a Ti:Sapphire regenerative amplifier (Spitfire-Ace PM1K, Spectra-Physics, CA, USA) seeded by a Ti:Sapphire solid-state laser (Mai Tai HP, Spectra-Physics, USA) and pumped by a diode-pumped Nd:YLF laser (Ascend 40, Spectra-Physics, USA). A homebuilt optical setup ensured control over the laser fluence at the sample plane by adjusting the pulse energy. Scanning of the 40 µm diameter laser beam over the substrate was achieved by implementing galvo-mirrors in the optical setup. Sufficient overlap between neighboring spots was provided to ensure that every location within the scanned area received at least one laser pulse (or two in the overlapping areas). To avoid field curvature aberrations induced by the optical setup, galvanometric scanning was limited to an area of 1 mm × 1 mm. Treatment of larger areas was achieved by tiling these scanned areas by moving the substrate using a motorized XY scanning stage (Thorlabs, Newton, NJ, USA).

For adherent cells (HeLa cells in this work), 15,000 cells were seeded in one well of a 96-well plate 24 h before photoporation treatment. After the incubation with the graphene-based material, free particles are washed away once by DPBS. Before laser treatment, FITC-dextrans (FDs) of different molecular weights (1 mg/mL 10 kDa (FD10), 70 kDa (FD70) or 500 kDa (FD500)) diluted in cell culture medium were added into the wells. After laser scanning, the cell medium with FITC-dextran was immediately removed to avoid further uptake by endocytosis. The cells were gently washed with DPBS and supplemented with fresh cell culture medium (CCM).

For suspension cells (Jurkat cells in this work were obtained from the American Type Culture Collection, ATCC® TIB-152®), 250,000 cells were suspended in one well of a 96-well plate. After incubation with the graphene-based materials, cells were collected by centrifugation and then re-suspended in CCM. Another centrifugation was performed to wash unbound nanoparticles. Before laser treatment, Jurkat cells were re-suspended in FD10 solution (1 mg/mL in CCM). The laser treatment was performed after 5 min, which was sufficient to let the cells sediment on the bottom of the substrate. Afterward, the cells were immediately collected, centrifuged, and washed three times with fresh CCM, and finally put on fresh CCM again.

### 3.8. VNB Generation and Visualization in Living Cells

The VNB generation and visualization were done on the 561-nm laser setup. For VNB threshold determination, the graphene materials are dispersed in DPBS and deposited on coverslips. The particles were irradiated by 7 ns laser pulses with a wavelength of 561 nm. VNBs are visualized under dark-field microscopy. The laser fluence threshold can be determined by counting the number of visible VNBs within the laser spot (150 µm laser beam diameter) for increasing laser pulse energies. The VNB threshold was defined as the laser fluence at which VNBs were formed with 90% certainty [17]. Then the number of VNBs was performed as a function of laser fluence with a Boltzmann fit. The function form is Y = Bottom+(Top−Bottom)/(1 + exp((V50–X)/Slope)), where “Top”, “Bottom”, “Slope” and “V50” are constants that are calculated by the fitting. “Top” is the maximum number of VNBs and “X” is the VNB formation threshold that needs to be determined. All further experiments were performed at approximately twice the VNB threshold to be certain that VNBs were effectively formed.

To visualize VNBs in cells, 60,000 HeLa cells were seeded on a 50 mm glass-bottom dish (MatTek Corporation, Ashland, MA, USA) one day in advance. The cells were incubated for 30 min with 8.9 × 10^8^ nps/mL GQD, 3.4 × 10^9^ nps/mL GQD-PEG, 1.31 × 10^10^ nps/mL rGO, 2 × 10^9^ nps/mL rGO-PEG, and 3.2 × 10^8^ nps/mL rGO-PEI. Cells were washed once with DPBS after the incubation and then supplemented with fresh CCM. An electronic pulse generator (BNC575, Berkeley Nucleonics Corporation, San Rafael, CA, USA) was used to generate single laser pulses and trigger the camera (EMCCD camera, Cascade II: 512, Teledyne Photometrics, Tucson, AZ, USA) to record images before and during VNB generation. The laser pulse energy was measured by an energy meter (J-25MB-HE and LE, Energy Max-USB/RS sensors, Coherent, Menlo Par, CA, USA).

### 3.9. Confocal Microscopy Imaging

Confocal microscopy imaging was performed on a laser scanning confocal microscope (C2, Nikon, Tokoyo, Japan) using a 10× magnification (CFI Plan Apo VC, Nikon, Japan). The 488 nm continuous wave laser (Coherent Sapphire, Menlo Par, CA, USA) was used for excitation of FITC-dextran.

### 3.10. Flow Cytometry

Following photoporation with FD10 and performing several washing steps as described above, HeLa cells were detached by trypsin-EDTA treatment and collected by centrifugation. Jurkat cells were immediately ready for flow cytometry following the washing steps, as described above. Cells were re-suspended in flow buffer (DPBS supplemented with 1% *w/v* BSA and 0.1% *w/v* NaN_3_) and the samples were measured by flow cytometry with 96-well plate loader (CytoFLEX, Beckman Coulter, Krefeld, Germany). Eight thousand cells were measured per sample. A 488-nm laser was used to excite FITC-dextrans and the fluorescence intensity was recorded in the 530/30 channel. All data are expressed as the mean ± SD (*n* = 3).

### 3.11. Cell Viability Assay by CellTiter Glo^®^

After the photoporation and washing step, cells were put back in the incubator to allow for an initial recovery for 2 h. Afterward, the HeLa cell medium was replaced by 100 µL fresh CCM. For Jurkat cells, they were centrifuged down and then re-suspended in 100 µL fresh CCM. One hundred microliters of CellTiter Glo^®^ solution (CellTiter-Glo^®^ Luminescent Cell Viability Assay, Promega, WI, USA) was added into each well. The plate was put on the shaker at 100 rpm for 10 min. The luminescence was measured by GloMax (GloMax^®^ 96 Microplate Luminometer, Promega). All data are expressed as the mean ± SD (*n* = 3).

## 4. Conclusions

We have demonstrated that the functionalization of GQD and rGO with PEG or PEI improves their performance for the intracellular delivery of macromolecules into adherent and suspension cells by photoporation. By enhancing their colloidal stability, their photothermal effect on cells is more uniform, which turns out to improve transfection efficiency and viability. PEI functionalization proves to be the most beneficial, likely due to stronger binding to the negatively charged cell membrane as compared to PEG functionalization. Together these results demonstrate that with a proper surface functionalization, graphene-based nanomaterials represent an interesting class of sensitizing nanoparticles for efficient cell transfections by photoporation. rGO, in particular, offers the benefit of being excitable by a broad range of laser wavelengths, including NIR irradiation, which is expected to be beneficial for transfection in thicker biological tissues.

## Figures and Tables

**Figure 1 ijms-21-01540-f001:**
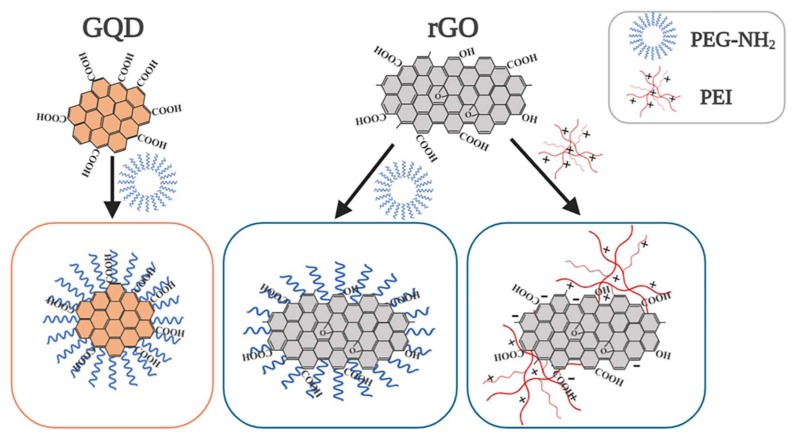
Schematic illustration of the modification of graphene quantum dots (GQD) and reduced graphene oxide (rGO). Amino-polyethylene glycol (PEG) is covalently coupled to carboxyl or hydroxyl groups present on GQD and rGO, while polyethyleneimine (PEI) is electrostatically adsorbed to rGO. “+” and “−” stand for positive and negative charge, respectively. Created with Biorender.com.

**Figure 2 ijms-21-01540-f002:**
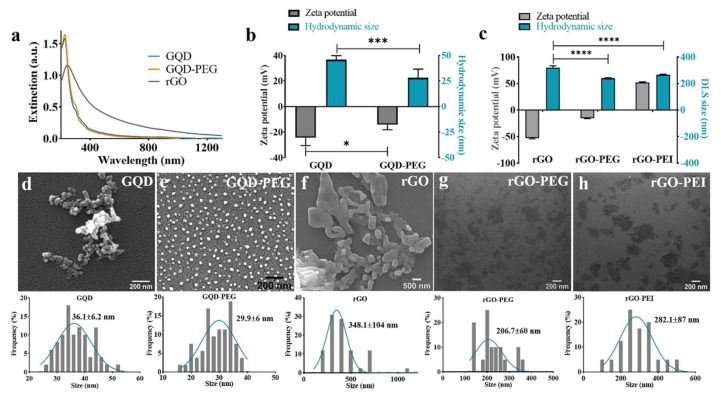
Characterization of physicochemical properties of GQD and rGO before and after functionalization with PEG or PEI. (**a**) The UV-NIR distinction spectra of GQD, GQD-PEG, and rGO. (**b**) Zeta potential and hydrodynamic size of GQD and GQD-PEG as measured by DLS in deionized water. (**c**) Zeta potential and hydrodynamic size of rGO, rGO-PEG, and rGO-PEI as measured by DLS in deionized water. SEM images and size distributions of (**d**) GQD, (**e**) GQD-PEG, (**f**) rGO, (**g**) rGO-PEG, and **h)** rGO-PEI. A gold layer was applied to the GQD, GQD-PEG and rGO samples. while the rGO-PEG and rGO-PEI were directly imaged on a silicon wafer. * *p* < 0.1, **** *p* < 0.0001, and ns = not significant.

**Figure 3 ijms-21-01540-f003:**
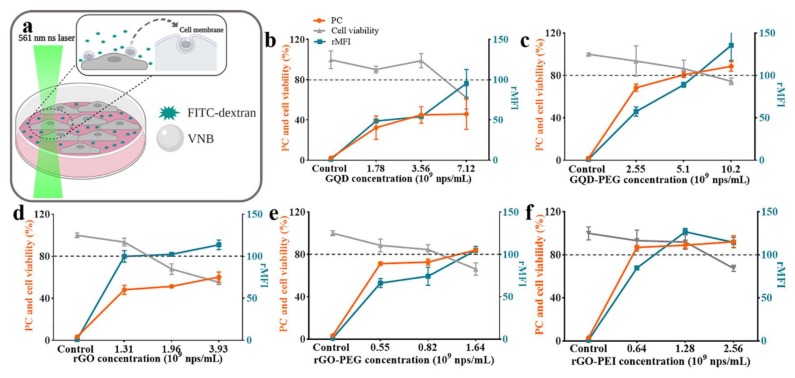
FD10 delivered into living HeLa cells by photoporation with graphene-based materials. (**a**) Conceptual illustration of the photoporation experiment. Created with Biorender.com. (**b**–**f**). Quantification by flow cytometry of the percentage of FD10 positive cells (PC) and relative mean fluorescence intensity (rMFI) (MFI compared to control cells). Cell viability was determined by CellTiter-Glo^®^ 2 h after the photoporation. The same protocol was applied to control cells but in the absence of nanoparticles. (**b**) GQD, (**c**) GQD-PEG, (**d**) rGO, (**e**) rGO-PEG, and (**f**) rGO-PEI.

**Figure 4 ijms-21-01540-f004:**
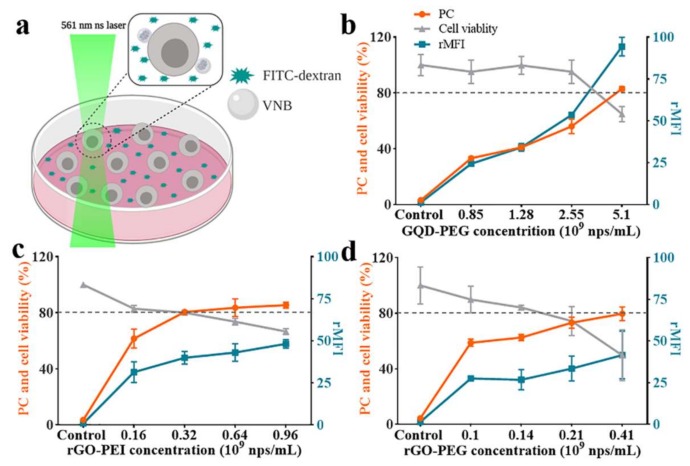
FD10 delivered into Jurkat cells by photoporation with polymer-modified graphene-based materials. (**a**) Conceptual illustration of the photoporation experiment. Created with Biorender.com. (**b**–**d**) Quantification by flow cytometry of the percentage of FD10 positive cells (PC) and rMFI (MFI compared to control cells). Cell viability is determined by CellTiter-Glo^®^ 2 h after the photoporation. The same protocol is applied to control cells but in the absence of nanoparticles. (**b**) GQD-PEG, (**c**) rGO-PEG, and (**d**) rGO-PEI.

**Figure 5 ijms-21-01540-f005:**
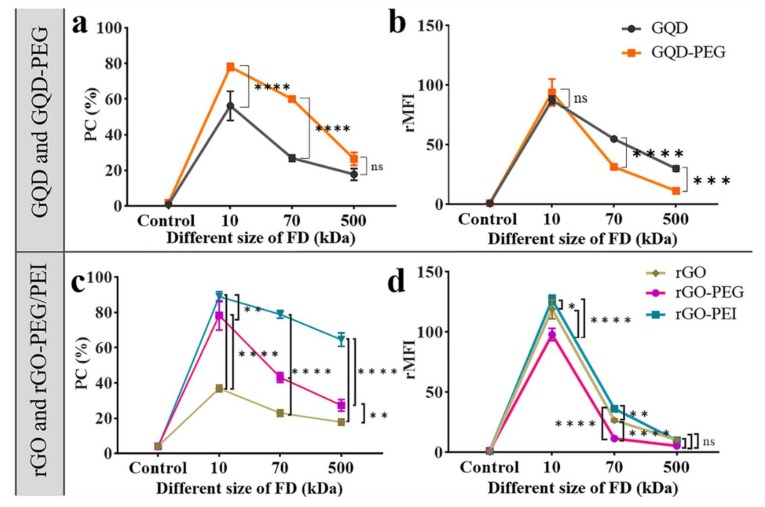
Flow cytometry results of HeLa cells photoporated with FD10, FD70, and FD500. (**a**) and (**b**) HeLa cells are photoporated using 3.56 × 10^9^ nps/mL GQD and 5.1 × 10^9^ nps/mL GQD-PEG, respectively. (**c**) and (**d**) HeLa cells were photoporated using 1.31 × 10^9^ nps/mL rGO, 0.82 × 10^9^ nps/mL rGO-PEG, and 12.8 × 10^8^ nps/mL rGO-PEI, respectively. (**a**) and (**c**) Positive cells were quantified by flow cytometry. (**b**) and (**d**) rMFI of positive cells. **** *p* < 0.0001, *** *p* < 0.001, ** *p* < 0.01, and ns = not significant.

**Figure 6 ijms-21-01540-f006:**
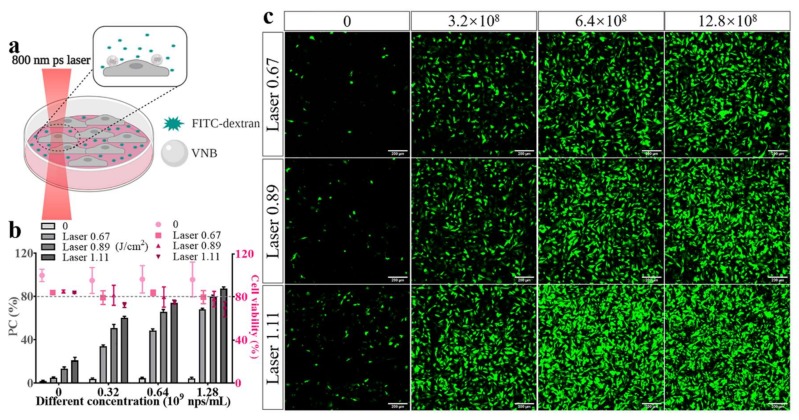
Intracellular delivery of FD10 by NIR laser-induced photoporation with rGO-PEI. (**a**) Conceptual illustration of the photoporation experiment. Created with Biorender.com. (**b**) Quantification of the percentage of FD10 positive cells by flow cytometry. Cell viability was determined by CellTiter-Glo^®^ 2 h after photoporation treatment. (**c**) Confocal microscopy images of HeLa cells labeled with FD10 with increasing particle concentrations (columns) and increasing laser fluence (rows). The scale bar is 200 µm.

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
