# Peer review of "Surface Functionalization with Polyethylene Glycol and Polyethyleneimine Improves the Performance of Graphene-Based Materials for Safe and Efficient Intracellular Delivery by Laser-Induced Photoporation"

_ijms, 2020, doi:10.3390/ijms21041540_

Round 1

Reviewer 1 Report

In this work the authors functionalized graphene quantum dots and reduced graphene oxide with polyethylene glycol and  polyethyleneimine to enhance the colloidal stability of QDs while
retaining their photoporation functionality.
The article is well written and the data presented satisfactorily.
Before publishing I would like to suggest some small changes to the authors.

-authors should indicate keywords at the end of the abstract;

-some acronyms are defined in the abstract, I suggest defining them again in the text the first time they are used to favor non-expert readers;

-I suggest checking some typos (such as in the caption in fig. 6), and simplifying sentences that are too long (for example at the beginning of the introduction);

-why for rGO the size significantly decreased after PEGylation? The authors should provide some hypotheses.

-The degree of aggregation that is carried out dry on the TEM grid is obviously different from that which is achieved in solution. For this the authors should spend a few more words on DLS data;

the authors say that in previous work they observed aggregation of quantum dots in biological buffers. Do they have any data relating to the pegylated QDs obtained in this work? I am referring in particular to DLS data. These could better support the arguments made in the text;

-I suggest eliminating the separation between the "results" and "discussion" paragraphs as the authors did not just expose the results but commented them gradually. I would simply eliminate the distinction between the two paragraphs leaving the text unchanged.

-the authors should also show the absorption spectra of the pegylated QDs in figure 2a. Are the spectra shown in figure 2a correct for the scattering? A scattering contribution can be seen in the spectrum of the rGOs which are around 350 nm in size. Before comparing the absorption of these rGOs with that of the GQD, it is necessary to eliminate this contribution which is all the more significant in the ultraviolet region. 

Author Response

Rebuttal letter to reviewer

First of all, we would like to thank all reviewers for the careful reading of our manuscript and providing useful comments and suggestions. We have considered them carefully and adjusted the manuscript accordingly. We have also performed additional experiments to further substantiate our main conclusions. We hope that the reviewers will agree that this has further strengthened the manuscript.

Comments from the reviewer #1:

In this work the authors functionalized graphene quantum dots and reduced graphene oxide with polyethylene glycol and polyethyleneimine to enhance the colloidal stability of QDs while retaining their photoporation functionality. The article is well written and the data presented satisfactorily.

Before publishing I would like to suggest some small changes to the authors.

  1. authors should indicate keywords at the end of the abstract;

RESPONSE:

According to the reviewer’s comments, we have added the following key words: “photoporation; intracellular delivery; nanomaterial functionalization; graphene-based materials; colloidal stability”.

  1. some acronyms are defined in the abstract, I suggest defining them again in the text the first time they are used to favor non-expert readers;

RESPONSE:

According to the reviewer’s suggestion, we have also defined the acronyms in the text.

  1. I suggest checking some typos (such as in the caption in fig. 6), and simplifying sentences that are too long (for example at the beginning of the introduction);

RESPONSE:

We have carefully reread the manuscript and corrected any typos we could find, which are marked in the text.

  1. why for rGO the size significantly decreased after PEGylation? The authors should provide some hypotheses.

RESPONSE:

Actually we notice that the hydrodynamic size of graphene-based nanomaterials decreases after functionalization, both for GQD and rGO (Figure 2b and 2c). To be certain of this we have done new DLS experiments, again confirming that the hydrodynamic size (number mean) of GQD in deionized water (45.8 nm) is larger than for GQD-PEG (28.2 nm). This decrease in size was also observed in the SEM images. Similarly, for rGO the hydrodynamic size (number mean) significantly decreases after functionalization from 320.4 nm to 240.1 nm for rGO-PEG, and to 266.2 nm for rGO-PEI, as was found by SEM as well. We attribute this to the better colloidal stability after surface functionalization, meaning that GQD-PEG, rGO-PEG and rGO-PEI are more monodispersed as compared to the non-functionalized nanomaterials. We have updated Figure 2 with the new DLS data and included a statement on the improved colloidal stability in the section “Physicochemical characterization of GQD and rGO before and after surface functionalization”.

Figure 2. Characterization of physicochemical properties of GQD and rGO before and after functionalization with PEG or PEI. a. The UV-NIR absorption spectra of GQD, GQD-PEG and rGO. b. Zeta potential and hydrodynamic size of GQD and GQD-PEG as measured by DLS. c. Zeta potential and hydrodynamic size of rGO, rGO-PEG, and rGO-PEI as measured by DLS. SEM images and size distributions of d. GQD, e. GQD-PEG, f. rGO, g. rGO-PEG and h. rGO-PEI. A gold layer was applied to the GQD, GQD-PEG and rGO samples. while the rGO-PEG and rGO-PEI were directly imaged on a silicon wafer. *P < 0.1, ****P < 0.0001, and ns = not significant.

  1. The degree of aggregation that is carried out dry on the TEM grid is obviously different from that which is achieved in solution. For this the authors should spend a few more words on DLS data;

RESPONSE:

Since both techniques measure nanoparticle size in very different ways the size values determined by DLS and SEM are not exactly identical. DLS calculates an equivalent spherical hydrodynamic diameter, while SEM sizes are based on direct visualization and image analysis. Nevertheless the average size values as determined by SEM and DLS are very close. Importantly, both techniques give consistent results with regard to a decreasing size after surface functionalization, as discussed in the previous comment.

  1. the authors say that in previous work they observed aggregation of quantum dots in biological buffers. Do they have any data relating to the pegylated QDs obtained in this work? I am referring in particular to DLS data. These could better support the arguments made in the text;

RESPONSE:

We performed additional size measurements of GQD and GQD-PEG in HEPES buffer (instead of deionized water as before). As shown in Figure S1a, the hydrodynamic size of GQD increased to 101 nm while this was not the case for GQD-PEG (44.2 nm). This clearly indicates the aggregation of GQD. These data are added to Figure S1 and discussed in the section “Physicochemical characterization of GQD and rGO before and after surface functionalization”.

Figure S1. Raman spectra of GQD and GQD-PEG. a. Hydrodynamic size of GQD, and GQD-PEG as measured by DLS in HEPES buffer. b. Raman spectra of GQD and GQD-PEG and c. of rGO, rGO-PEG, and rGO-PEI. The dashed lines mark the position of the D (1326 cm-1) and G bands (1610 cm-1). d. Quantification of the ID/IG ratio. A statistical difference is found between GQD and GQD-PEG, rGO and rGO-PEG, and rGO and rGO-PEI, confirming successful modification of GQD with PEG and rGO with PEG and PEI. *P < 0.01, ***P < 0.001.

  1. I suggest eliminating the separation between the "results" and "discussion" paragraphs as the authors did not just expose the results but commented them gradually. I would simply eliminate the distinction between the two paragraphs leaving the text unchanged.

RESPONSE:

Based on the reviewer’s suggestion we deleted the title ‘discussion’ and renamed ‘results’ to ‘results and discussion’.

  1. the authors should also show the absorption spectra of the pegylated QDs in figure 2a. Are the spectra shown in figure 2a correct for the scattering? A scattering contribution can be seen in the spectrum of the rGOs which are around 350 nm in size. Before comparing the absorption of these rGOs with that of the GQD, it is necessary to eliminate this contribution which is all the more significant in the ultraviolet region.

RESPONSE:

As suggested by the reviewer, we have added the absorption spectrum of pegylated GQDs in Figure 2a. Since there is almost no absorption of the PEG in the range of 200 nm to the NIR wavelength, the absorption spectrum of GQD-PEG is almost the same as that of GQD.

Actually the spectra we showed in this figure are the extinction spectra in the UV-NIR range, measured by NanoDrop. As such they are not corrected for scattering. For the purpose of this manuscript this doesn’t matter much since we excite the particles in the visible (561 nm) and NIR (800 nm) range. In the revised manuscript we changed “absorption spectra” to “extinction spectra” to be more correct in this regard.

Figure 2. Characterization of physicochemical properties of GQD and rGO before and after functionalization with PEG or PEI. a. The UV-NIR absorption spectra of GQD, GQD-PEG and rGO. b. Zeta potential and hydrodynamic size of GQD and GQD-PEG as measured by DLS. c. Zeta potential and hydrodynamic size of rGO, rGO-PEG, and rGO-PEI as measured by DLS. SEM images and size distributions of d. GQD, e. GQD-PEG, f. rGO, g. rGO-PEG and h. rGO-PEI. A gold layer was applied to the GQD, GQD-PEG and rGO samples. while the rGO-PEG and rGO-PEI were directly imaged on a silicon wafer. *P < 0.1, ****P < 0.0001, and ns = not significant.

Reviewer 2 Report

The authors present a study on how to improve photoporation efficiency using graphene-based nanomaterials by decorating their surfaces with polymers. The conclusions follow logically from the results and the article is very clearly written.

I have only a few issues with the manuscript, all of which I think can easily be addressed and do not affect the scientific conclusions:

It is not clear to me what a “Boltzmann fit” is (e.g., Fig. S3 and associated description in the main text). Some clarification (including the functional form) would be useful.

It is also not clear to me how the concentrations (number of nanoparticles per unit volume) were estimated. I presume the numbers reported are approximations (possibly based on the known total mass and using the DLS sizes) but how this was performed could be included (briefly).

Minor comments:

I think “cytoplasm” should be “cytosol” throughout the article, unless the authors really intend the former (which would include organelles).

Figure 3: The (left) y axis of panels b-f should not extend (much) beyond 100%, since there are no values above 100%.

Figure S2: The figure caption denotes the rows incorrectly (GQD is not included so all the row numbers should be increased by one).

Figure S7: The figure caption describes a panel f, but no such panel is included.

In terms of language, there are only a few mistakes with one exception: The tense occasionally changes from the past tense to present tense (and even future tense), with no justification. There is no issue with the language being ambiguous, but the review report template does ask for English language and style and hence I note it.

Author Response

Rebuttal letter to reviewer

First of all, we would like to thank all reviewers for the careful reading of our manuscript and providing useful comments and suggestions. We have considered them carefully and adjusted the manuscript accordingly. We have also performed additional experiments to further substantiate our main conclusions. We hope that the reviewers will agree that this has further strengthened the manuscript.

Comments from the reviewer #2:

The authors present a study on how to improve photoporation efficiency using graphene-based nanomaterials by decorating their surfaces with polymers. The conclusions follow logically from the results and the article is very clearly written.

I have only a few issues with the manuscript, all of which I think can easily be addressed and do not affect the scientific conclusions:

It is not clear to me what a “Boltzmann fit” is (e.g., Fig. S3 and associated description in the main text). Some clarification (including the functional form) would be useful.

RESPONSE:

In this manuscript, the “Boltzmann fit” is automatically processed in GraphPad with a function form of Y=Bottom+(Top-Bottom)/(1+exp((V50-X)/Slope)), where “Top”, “Bottom”, “Slope” and “V50” are constants that are calculated by the fitting. “Top” is the maximum number of VNBs and “X” is the VNB formation threshold that needs to be determined.

To make this fitting method clearer, we have added this equation and explanation to the Experimental section of “VNB generation and visualization in living cells”.

It is also not clear to me how the concentrations (number of nanoparticles per unit volume) were estimated. I presume the numbers reported are approximations (possibly based on the known total mass and using the DLS sizes) but how this was performed could be included (briefly).

RESPONSE:

The concentration of the graphene-based materials are determined by Nanoparticle Tracking Analysis (NanoSight LM10, Malvern, UK), which was mentioned in the Experimental section. To make it clearer to the readers, we have also added this information in section 3.2.

Minor comments:

I think “cytoplasm” should be “cytosol” throughout the article, unless the authors really intend the former (which would include organelles).

RESPONSE:

This is correct and we have changed “cytoplasm” to “cytosol” to avoid confusion.

Figure 3: The (left) y axis of panels b-f should not extend (much) beyond 100%, since there are no values above 100%.

RESPONSE:

We have adjusted the (left) y axis of Figure 3 and Figure 4, as shown below.

Figure 3. FD10 delivered into living HeLa cells by photoporation with graphene-based materials. a. Conceptual illustration of the photoporation experiment. Created with Biorender.com. b-f. Quantification by flow cytometry of the percentage of FD10 positive cells (PC) and rMFI (= MFI compared to control cells). Cell viability was determined by CellTiter-Glo® 2h after the photoporation. The same protocol was applied to control cells, but in the absence of nanoparticles. b. GQD, c. GQD-PEG, d. rGO, e. rGO-PEG, f. rGO-PEI.

Figure 4. FD10 delivered into Jurkat cells by photoporation with polymer-modified graphene-based materials. a. Conceptual illustration of the photoporation experiment. Created with Biorender.com. b-d. Quantification by flow cytometry of the percentage of FD10 positive cells (PC) and rMFI (= MFI compared to control cells). Cell viability was determined by CellTiter-Glo® 2h after the photoporation. The same protocol was applied to control cells, but in the absence of nanoparticles. b. GQD-PEG, c. rGO-PEG, d. rGO-PEI.

Figure S2: The figure caption denotes the rows incorrectly (GQD is not included so all the row numbers should be increased by one).

RESPONSE:

Maybe there is some mistake in the previous version, but the GQD is included in the figure now.

Figure S7: The figure caption describes a panel f, but no such panel is included.

RESPONSE:

This mistake is now corrected.

In terms of language, there are only a few mistakes with one exception: The tense occasionally changes from the past tense to present tense (and even future tense), with no justification. There is no issue with the language being ambiguous, but the review report template does ask for English language and style and hence I note it.

RESPONSE:

We have carefully gone through the manuscript and adjusted the tense as needed. These words are marked in the manuscript.
